# A Journey to Reach the Ovary Using Next-Generation Technologies

**DOI:** 10.3390/ijms242316593

**Published:** 2023-11-22

**Authors:** Thuy Truong An Nguyen, Isabelle Demeestere

**Affiliations:** Research Laboratory on Human Reproduction, Faculty of Medicine, Université Libre de Bruxelles (ULB), 1070 Brussels, Belgium; thuy.nguyen@ulb.be

**Keywords:** gold nanoparticles, microRNA therapies, ovaries, fertility preservation

## Abstract

Although effective in terms of the chances of future live birth, the current methods for fertility preservation, such as oocyte, embryo, or ovarian tissue cryopreservation, cannot be offered to all cancer patients in all clinical contexts. Expanding options for fertility preservation is crucial to addressing the need to encompass all situations. One emerging strategy is pharmacoprotection, a non-invasive approach that has the potential to fill existing gaps in fertility preservation. In addition to the identification of the most effective therapeutic agents, the potential for off-target effects remains one of the main limitations of this strategy for clinical application, particularly when healthy ovarian tissue is targeted. This review focuses on the advances in pharmacoprotective approaches and the challenge of targeting the ovaries to deliver these agents. The unique properties of gold nanoparticles (AuNPs) make them an attractive candidate for this purpose. We discuss how AuNPs meet many of the requirements for an ideal drug delivery system, as well as the existing limitations that have hindered the progression of AuNP research into more clinical trials. Additionally, the review highlights microRNA (miRNA) therapy as a next-generation approach to address the issues of fertility preservation and discusses the obstacles that currently impede its clinical availability.

## 1. Introduction

Fertility preservation (FP) for female cancer patients should be considered before initiating chemotherapy, radiotherapy, or other anti-cancer therapies. Unfortunately, several chemotherapeutic agents and radiation exposure have side effects, including potential long-term impacts on fertility, by inducing premature ovarian insufficiency (POI). It is crucial to improve the quality of life of these patients after recovery from cancer, given that five-year cancer survival rates have increased in recent decades [1] due to constant advances in cancer treatment and personalized therapeutic medicine. Cryopreservation of oocytes [2], embryos [3], or ovarian tissue [4] is considered to be the standard method for FP. Although these techniques have been proven to be effective, they are invasive and have limitations such as the risk of reinsertion of cancer cells after transplantation of cryopreserved ovarian tissue, the only option available for pre-pubertal cancer patients [5].

Interest has arisen in alternative methods for FP to overcome the limitations of current techniques, including ovarian tissue engineering [6], in vitro ovarian tissue reconstitution through stem cell differentiation [7], and pharmacoprotective agents. The main advantage of this last approach is that it is non-invasive. This aspect has led to the study of numerous molecules to investigate their protective effects on fertility. These molecules can be categorized into two classes: those aimed at protecting against DNA damage [8], which induces apoptosis in ovarian follicles, and those aimed at protecting against the burn-out effect [9], characterized by excessive activation of primordial follicles. Both phenomena lead to POI by reducing the ovarian reserve. Anti-apoptotic molecules include imatinib, GNF-2, and sphingosine-1-phosphate (S1P). Imatinib and GNF-2 are ABL kinase inhibitors and interfere with the process of DNA damage that ultimately leads to TAp63 activation and, finally, apoptosis [10,11]. Studies have demonstrated protection of the ovarian reserve against cisplatin- and cyclophosphamide-induced apoptosis in mouse models by imatinib and GNF-2, respectively. S1P is an inhibitor of ceramide-induced death pathways and demonstrates the prevention of ovarian follicle apoptosis after in vitro exposure to cyclophosphamide and doxorubicin in mouse and human models [12].

The second main mechanism of follicular depletion after chemotherapy exposure is called the “burn-out” effect. It is defined by a massive activation of the quiescent follicles due to direct damage to growing follicles, leading to a dramatic reduction of a key regulator of follicular activation, anti-Müllerian hormone (AMH), and activation of the PI3K-AKT-mTOR pathway, inducing granulosa cell (GC) proliferation [13,14]. Examples of molecules that have been studied to prevent the burn-out effect include AS101, rapamycin, and ghrelin, which all act on the PI3K-AKT-mTOR pathway and are known to be involved in follicle activation. These agents are inhibitors of PI3K, mTOR, and FOXO3a, respectively [15,16,17].

The most advanced research among protective molecules has focused on gonadotropin-releasing hormone agonists (GnRHa). While the above-mentioned molecules are in pre-clinical studies, GnRHa is currently used in clinical practice, although its efficacy remains controversial. The main mechanism of action that has been proposed is that co-treatment with GnRHa and a chemotherapeutic agent maintains low levels of follicle-stimulating hormone (FSH) to keep follicles at the quiescent stage, as in the pre-pubertal hormonal environment. However, this hypothesis is still not fully understood, considering the fact that follicular activation is gonadotropin-independent [18]. Additionally, studies have yielded opposing results depending on the type of cancer treated. While breast cancer patients seem to benefit from co-treatment with a better recovery of ovarian function [19], this advantage is not observed in lymphoma patients [20].

Among innovative ovarian protective approaches, microRNA (miRNA) therapies represent another emerging strategy for FP and can be classified as pharmacoprotective agents due to their mechanism of action on gene expression [21]. The goal is to inhibit or restore the expression of specific mRNAs that have been identified as either overexpressed or downregulated and demonstrated to play a role in chemotherapy-induced ovarian damage. However, this growing interest and research in protective molecules raises the question of how to deliver these chemical agents. Finding the right delivery system to target the ovary could open a new frontier in reproductive medicine and FP treatment. The ideal delivery system should meet the following requirements: (1) Safety and stability: the agent does not induce toxicity or immunogenicity to allow circulation in the blood without degradation by nucleases or detection by the immune system; (2) Specificity: the agent is capable of reaching the targeted tissue by controlling renal clearance while being an appropriate size to cross the capillary endothelium; (3) Cellular uptake: the agent is able to enter cellular targets, penetrate the cell membrane, and reach the correct intracellular site of action; (4) Release: the agent is capable of delivering the therapeutic molecule, escaping the endolysosomal system, and avoiding exocytosis before the initiation of the action of the therapeutic molecule; (5) Efficacy: the agent is capable of carrying the necessary amount of payload that allows the therapeutic molecule to initiate its action and function; (6) Deliverability: the agent is feasible to manufacture with scalable and affordable possibilities.

Current delivery systems that have been studied are typically classified into two categories: viral and non-viral carriers. Viral vectors, including lentiviral, adenoviral, and adeno-associated viral (AAV) vectors, are already employed in clinical applications, mainly for virus vaccines, due to their inherent high delivery efficiency [22]. Unfortunately, pre-existing immunity toward these viruses can exist in humans, and achieving organ specificity apart from their initial tissue affinity can be challenging. The non-viral category includes systems based on lipidic, inorganic materials (such as silver, porous silica, and gold), polymeric molecules (including chitosan, PEI, and PAMAM), or extracellular vesicles (exosomes) (Figure 1). Among these non-viral vectors, the most advanced one for nucleic acid delivery, in terms of administration to a broad range of individuals, is the lipid nanoparticle (LNP), as demonstrated by the mRNA COVID-19 vaccine technologies developed and commercialized by Moderna Therapeutics (Cambridge, MA, USA) and Pfizer/BioNtech (New York, NY, USA/Mainz, Germany) [23]. Although this recent outbreak has opened the door for RNA therapeutics, achieving specificity through active targeting remains a challenge. Among the non-viral vectors mentioned, gold nanoparticles (AuNPs) stand out as one of the most tunable systems, a required property for the functionalization of the carrier to overcome various biological obstacles, in addition to their minimal toxicity and cost-effective synthesis.

This review will explore how AuNPs can meet these expectations and why they are a strong candidate for drug delivery of protective molecules, particularly miRNA therapies. In addition, the advantages, current applications, limitations, and gaps in this field will be highlighted. Finally, the review will focus on the potential role of miRNA therapy as part of the next generation of medicines.

## 2. Gold Nanoparticles as Next-Generation Delivery Systems

AuNPs are interesting due to their biocompatibility properties, their inert nature, and their high ability to be tunable in size, shape, and charge. Their most useful characteristic is their capacity to be multi-functionalized by binding molecules that provide the desired function (Figure 1). In fact, their surface chemistry is relatively easy to modify due to the quantity of functional groups that have an affinity for gold, such as carboxyls, thiols, and amines [24]. Based on this property, any molecule that binds these functional groups can theoretically be loaded onto the surface of the AuNPs. The assembly of different bioconjugations on the same AuNP is possible due to their high surface-volume ratio. Thus, in order to add stability to the system and hide it from the immune system, different polymers can be attached to AuNPs. One example of this is polyethylene glycol (PEG), which is extensively used for this purpose, including in the severe acute respiratory syndrome coronavirus (SARS-CoV) vaccine on the surface of LNPs. PEG coating has the ability to increase the half-life time of AuNPs in blood by providing a steric barrier to reduce their interaction with plasma proteins, known as opsonization, and thus avoid their recognition and uptake by the reticuloendothelial system (RES), leading to phagocytic clearance [25].

As a delivery system, AuNPs can carry a large variety of cargos, including proteins, drugs, and nucleic acids, either individually or in co-delivery. With regard to peptide delivery, some AuNP constructions are in clinical trials, including CYT-6091 (phase I, started in 2006) and C19-A3 AuNPs (Phase I, started in 2016). CYT-6091 is an AuNP functionalized with PEG that carries recombinant human tumor necrosis factor alpha (rhTNFa). This cytokine is known for its anti-tumor effects, and treatment with it prior to chemotherapy reduces tumor growth by a greater extent. In phase I of the clinical study, selected patients had various types of cancer, such as adenocarcinoma of the colon, pancreas, lung, and rectum, as well as ocular melanoma and ductal carcinoma of the breast. The AuNPs were well tolerated by the patients at the dose necessary to target the tumor through systemic administration [26]. Following these positive results, phase I was completed in 2009 and will lead to phase II, where a combination of CYT-6091 and chemotherapy is planned to enhance the anti-tumor effect. This makes CYT-6091 the first AuNP therapy to enter clinical trials and the most advanced one. C19-A3 AuNPs carry the human proinsulin peptide to improve antigen-specific immunotherapy for the treatment of type I diabetes. Phase I is still ongoing, but initial results demonstrated good tolerability after intradermal administration [27].

The delivery of drugs has been extensively studied for chemotherapeutic agents, including doxorubicin [28], cyclophosphamide [29], and cisplatin [30]. Studies of these classic agents delivered by AuNPs are all in the pre-clinical stage, while a combination of phytochemicals, including mangiferin and curcuma, conjugated to AuNPs has reached a pilot clinical investigation step to treat breast cancer. After demonstrating that this AuNP drug treatment, called Nano Swarna Bhasma (NSB), reduces the tumor size in SCID female mice bearing human breast cancer, treatment administered to six patients reported no side effects [31].

In the area of RNA therapeutics, where the treatment consists of the use of RNA for gene expression modulation, the delivery of nucleic acids is widely used. The first RNA interference (iRNA) therapy using AuNPs tested in humans is known as NU-0129. This system consists of a gold core carrying small interfering RNA (siRNA) coated with a protective layer of oligoethylene glycol. SiRNAs are short RNA sequences, ranging from 20 to 25 nucleotides, synthesized to perfectly match a fragment of the messenger RNA (mRNA) sequence targeted to downregulate the expression of a specific gene at the transcript level. The siRNA is delivered as a duplex with a passenger strand and a guide strand, which is taken into the RNA-Induced Silencing Complex (RISC) to degrade the targeted mRNA and silence the gene. In the case of NU-0129, the siRNA interferes with the expression of the oncogene *Bcl2Like12* (*Bcl2L12*) to treat patients with glioblastoma. NU-0129 was administered intravenously to eight patients and successfully crossed the blood-brain barrier to reach the tumor. A correlation was observed between the accumulation of NU-0129 and the downregulation of *Bcl2L12*, along with the upregulation of its target genes, caspase-3 (cleaved caspase 3) and p53, at the protein level. The treatment was well tolerated overall, although two severe adverse events, rated as grade 3 out of 5, were observed. These adverse events were considered to be a consequence of the oncologic treatment. Phase 0, completed in 2020, is expected to be followed by a larger cohort study, with a particular focus on examining the consequences of the long-term accumulation of gold. Notably, more than 40% of the total gold content remained detectable up to 174 days after the trial enrollment, following a single-dose administration [32]. Table 1 summarizes the AuNPs that have been used as drug delivery agents and reached clinical trials.

Finally, AuNPs functionalized with target molecules are essential for achieving specific organ delivery, especially when active targeting is required. Passive targeting refers to the natural route taken by a delivery system when administered in the body. This includes organs such as the liver, spleen, and kidney. Enhanced permeability and retention (EPR) effect develops following the high neovascularization of a tumor and the large gaps formed between endothelial cells that compose the capillary blood vessels. This effect can be exploited for passive drug delivery to tumors (Figure 2). However, it has been shown that the EPR effect is not always selective enough to avoid severe potential off-target toxicity when systemically administered, and this has led to death in a miRNA therapeutic phase 1 trial based on a pH-dependent delivery strategy [33]. Local injections can also be applied to target tissues when they are accessible, such as the skin. Active targeting is necessary to reach organs that are not easily accessible, such as the ovary, or to improve the specificity of a specific type of cell within a tumor. To make this possible, appropriate molecules can be attached to the AuNPs, such as peptides, antibodies, aptamers, or vitamins (Figure 2) [34]. Such precise targeting allows for a decrease in toxic side effects while enhancing therapeutic efficacy. In 2008, Patra et al. were the first to design an antibody-conjugated AuNP to target the overexpression of epidermal growth factor receptor (EGFR) in pancreatic adenocarcinoma, using cetuximab to deliver the anticancer drug gemcitabine [35]. Since then, Patra’s team has applied their AuNP-cetuximab to targeting ovarian cancer, which also overexpresses EGFR, by loading it with *p53* plasmid DNA to restore protein expression and reduce tumor progression in vivo [36]. The team also synthesized AuNPs functionalized with a bio-inspired fusion protein carrier to target human epidermal growth factor receptor-2 (HER2) for ovarian cancer and simultaneously co-deliver doxorubicin and siRNA against *erbB2*. In the three studies, they demonstrated higher accumulation of gold at the tumor site in xenograft mice compared to the liver, kidney, or spleen, along with a reduction in tumor volume compared with AuNPs that were not complexed with the antibody. They also showed better delivery with AuNPs compared to antibodies attached to the cargo without AuNPs [37]. Once the AuNPs reach their target cells, various mechanisms of internalization can occur, including clathrin-, caveolae-dependent, or independent endocytosis and macropinocytosis. The choice of mechanism depends on the functionalization and properties of the AuNP system [38].

In cases where AuNPs are trapped within the endolysosomal system and are unable to access the site of action, strategies for escaping this degradation system have been developed. The most widely employed is known as the proton sponge effect. This involves the use of a cationic molecule, such as polyethylenimine, to attract protons into the lysosome compartment. This mediates a change in pH, creating an acidic environment that triggers the disruption of the lysosomal membrane, allowing the release of the AuNPs into the cytoplasm [39]. Another interesting study highlighting the potential for co-delivery in gene therapy, along with an escape strategy, involved the technology called CRISPR-Cas9 for “clustered regularly interspaced short palindromic repeats and its associated protein 9." CRISPR-Cas9 is a revolutionary gene editing therapy that utilizes a prokaryote-derived immune system to correct mutations responsible for genetic diseases. This therapy employs the Cas9 ribonucleoprotein, the enzyme responsible for cutting the targeted gene, along with an RNA guide, similar to siRNA therapy, and a donor DNA to replace the mutated gene with the corrected sequence. Currently, the main delivery system for the CRISPR-Cas9 system is an AAV viral vector. However, this vector has limited gene-packaging capacity, necessitating the use of multiple viruses for efficient gene modulation. This is where AuNPs may play a role. In this study, they designed AuNPs with a dense multilayer of attached DNA molecules to link the gold core with the DNA donor, where Cas9 molecules are adsorbed with the RNA guide. The entire system was coated with the polymer poly(N-(N-(2-aminoethyl)-2-aminoethyl) aspartamide) (PAsp(DET)) to disrupt the endosomal membrane. This study demonstrated an absence of cellular uptake in HEK293 cells when the polymer was not present, compared to an 80% rate of internalization when the polymer was complexed with the AuNPs. In vivo results indicated gene editing induction without causing an increase in inflammatory plasma cytokines or a reduction in body weight [40].

The final obstacle in cargo delivery is failure to release the load and the inability to access the site of action. To address this challenge, research is ongoing to develop linker molecules that are cleavable by internal or external stimuli, such as changes in pH or exposure to light. For instance, a photolabile bond, such as o-Nitrobenzyl, was utilized in a study in which near-infrared (NIR) radiation applied to the AuNPs allowed the cleavage and release of 5-fluorouracil [41]. Further research is required to assess the efficacy of this concept for AuNPs in vivo. However, it is important to note that, for most of the studies summarized in this review and in the literature, this problem has not been systematically encountered.

In addition, other AuNP-based technologies in clinical trials have exploited the unique optic, electronic, and physicochemical properties of gold for diagnosis, treatment, imaging, and sensing. These distinctive optical properties are attributed to their surface plasmon resonance (SPR), a physical phenomenon in which light excites the electrons of gold, causing them to resonate, absorb, and reflect light. This SPR enables the detection and sensing of AuNPs and their utilization in photothermal therapy (PTT). In this therapy, AuNPs accumulate in the tumor, which is then irradiated with NIR radiation. AuNPs absorb the radiation and convert it into heat inside the tumor, leading to cell death. This therapy, also called photothermal ablation, is under investigation with several AuNPs in clinical trials, including AuroShell, a silica-gold NP coated with PEG, for the treatment of various solid tumors. Nanospectra Biosciences, Inc. (Houston, TX, USA) initiated a phase I trial in 2015 and described the safety of AuroShell with two adverse events—an allergic reaction and epigastric pain [42]. These safety results allowed the expansion of this study to 16 prostate cancer patients, demonstrating that 62.5% of the patients were free of cancer after 3 months and 87.5% after 12 months [43]. Following these encouraging results, two other pilot studies were conducted with head and neck cancer patients (NCT00848042) and lung cancer patients (NCT01679470). A second gold nanoshell was developed to treat atherosclerotic plaques, reporting significant regression of coronary atherosclerosis with an acceptable level of safety [44]. Another therapy using AuNPs and their SPR properties is photodynamic therapy (PDT). As the name suggests, this therapy relies on light to trigger a reaction in the AuNPs. In this case, a photosensitizer agent, such as porphyrin, is added. When excited by light, a transfer of energy will produce reactive oxygen species (ROS), inducing the apoptosis of cancer cells only in the presence of oxygen in the tissue. Many projects working on this therapy are still in the pre-clinical stage, as is the combination of PTT and PDT [45]. The electro-optical properties of AuNPs can also be exploited for imaging. Their high density enables AuNPs to serve as contrast agents and to absorb X-ray radiation used for photoimaging. The sensing application includes both physical properties to be detected and those to be functionalized [46]. Lastly, a gold nanocrystal in suspension, called CNM-Au8, has entered phase II clinical trials for the treatment of amyotrophic lateral sclerosis (ALS), a neurodegenerative disorder. This drinkable drug, suspended in a bicarbonate solution, uses the newly discovered catalytic property of gold to reduce disease-associated oxidative stress [47].

## 3. MicroRNA Therapy as an Innovative Ovarian Protection Approach

MiRNAs are short non-coding RNAs (ncRNAs), typically consisting of 20–25 nucleotides, similar in size to previously described siRNAs. Both types of ncRNAs modulate gene expression at the post-transcriptional level. However, there is a key distinction between them: siRNAs are artificially designed sequences that perfectly match specific mRNAs, whereas miRNAs are endogenous and have the ability to target the expression of multiple mRNAs [48]. This versatility arises from the imperfect interaction between miRNAs and mRNA, typically involving complementary base pairing between the seed sequence of the miRNA (the first 2 to 7 nucleotides in the 5′ region) and the 3′-UTR sequence of the target mRNA. The biogenesis of miRNAs starts with the transcription of a primary miRNA (pri-miRNA) from the genome by RNA polymerase II. Pri-miRNA is a double-stranded RNA with a hairpin loop, capped at the 5′ end, and a polyadenosine (polyA) tail at the 3′ end. The nuclear proteins Drosha and DGCR8 remove the cap and polyA tail, forming the precursor miRNA (pre-miRNA). The pre-miRNA is then exported to the cytoplasm by crossing the nuclear membrane protein Exportin-5. In the cytoplasm, the RNase III enzyme Dicer cleaves the hairpin structure, generating a miRNA duplex. The Argonaute protein unwinds this duplex, and the mature single-strand miRNA can now incorporate the RISC complex. This complex is responsible for inhibition of mRNA translation or induction of its degradation (Figure 3A) [49].

The discovery of the first miRNA, lin-4, dates back to 1993 in the nematode *Caenorhabditis elegans* [50]. Since then, thousands of miRNAs have been identified across species, showing high levels of conservation in both miRNA sequences and their target mRNA interactions [51]. An estimation based on a bioinformatic study suggested that the human genome contains around 2300 miRNAs, with 1115 of them identified in miRBase, a miRNA database [52]. In total, human miRNAs are capable of modulating the expression of up to 60% of protein-coding genes, resulting in a complex and powerful regulatory network. MiRNAs regulate genes involved in various cell processes, including cell division, proliferation, apoptosis, and metabolism. It is not surprising that dysregulation of a crucial miRNA can lead to pathologies such as neurological disorders, cardiovascular diseases, diabetes, fibrotic diseases, and, most notably, cancer. Their significant roles in both normal and pathological physiology make miRNAs interesting candidates for diagnosis as markers and in therapeutic applications. In the context of cancer, miRNAs can be classified into two categories: tumor-suppressor miRNAs and oncogenic miRNAs. Tumor-suppressor miRNAs have their expression reduced, while oncogenes are overexpressed. In cancer therapy, the use of miRNAs can involve the replacement of tumor-suppressor miRNAs using miRNA mimics or the inhibition of oncogenic miRNAs using anti-miRNAs (Figure 3B).

The growing interest in miRNA-based therapy is reflected in the number of clinical studies and their progress over the past decade (Table 2) [53,54,55,56]. In the initial trials, modifications to the molecular structure enabled direct injection of the drug without the need for a delivery system. For instance, a modification in the ribose of the nucleic acid, called Locked Nucleic Acid (LNA), acts to enhance RNA stability and protect the molecule from degradation. The most advanced miRNA therapy study using a delivery system is TargomiRs. This treatment for patients with recurrent malignant pleural mesothelioma entered phase I trials in 2014 [57]. The delivery system used to encapsulate the miR-16 mimic is a nanocell, a non-living vesicle derived from the asymmetric division of bacteria, called EnGenIC Dream Vector by the biotech company. MiR-16 targets genes such as *BCL2* and *JUN* that are involved in cancer progression. Additionally, the nanocells are coated with EGFR-specific antibodies on their surface to target the tumor overexpressing EGFR on mesothelioma cells. In pre-clinical studies, tumor growth inhibition was observed in mice when the nanocells delivering miR-16 were injected into the tail [58]. The phase I trial reported an acceptable safety profile, and further studies are planned to combine TargomiRs with chemotherapy. However, issues related to cardiac toxicity and other adverse events suggesting an immune reaction need to be addressed first. Another miRNA-based replacement therapy, which was the first to employ a delivery system and to enter clinical trials, also induced an immune reaction but was classified as a severe adverse event and had to stop at Phase I after four patient deaths were reported. In this study, a miR-34a mimic was delivered using a LNP (MRX34) to treat patients with advanced solid tumors [33]. To date, no miRNA therapy has reached the market, although siRNA therapies have received approval from the US Food and Drug Administration and/or the European Medicines Agency. Patisiran, a siRNA targeting *transthyretin* and delivered by LNPs, is among these approved therapies. It is used to treat hereditary transthyretin-mediated amyloidosis. However, the clinical utility of Patisiran is limited due to the requirement for steroids and antihistamines before siRNA treatment to prevent immune reactions [59]. These clinical studies and therapies highlight the importance of targeting specific tissues and understanding the targeted mRNAs of a particular miRNA. Improving the specificity of delivery systems remains an ongoing challenge, as already discussed. Research is still necessary to achieve the greatest specificity, but interest in this challenge is growing. Concerning the target mRNAs and their impact on molecular pathways, advancements in bioinformatics, which continuously develop new miRNA target prediction tools, should help to resolve this obstacle [60].

Cancer therapy can significantly impact the expression profiles of miRNAs, as shown by studies reporting significant changes in miRNA expression in cancer cells during and after chemotherapy exposure [61,62]. For instance, a study on breast cancer patients revealed upregulation of tumor suppressor miRNAs in patients who received neoadjuvant bevacizumab in combination with chemotherapeutic agents. Additionally, downregulation of oncogenic miRNAs was observed in patients treated with neoadjuvant therapy only. Importantly, these results were detected in responding patients. Among the downregulated oncogenic miRNAs, miR-4465 showed the strongest correlation with a reduction in tumor cell proliferation. Based on correlated and predicted target genes, miR-4465 was identified as a regulator of genes associated with cell cycle regulation and DNA damage response (DDR) [63]. Conversely, miRNAs can affect chemotherapy by either promoting chemoresistance [64] or enhancing chemosensitivity [65]. MiRNAs can play a role in chemoresistance by regulating genes involved in DDR, drug efflux pumps that maintain high drug concentrations inside the cells, and tumor survival mechanisms [66]. On the other hand, miRNAs can enhance chemosensitivity, as reported in a study where let-7a downregulated genes involved in metabolic reprogramming crucial for cell cycle progression. This study also demonstrated that let-7a can induce the production of ROS in breast cancer cells [67]. Let-7a is known as a tumor suppressor, as some of its target genes include oncogenes such as *MYC*, *RAS*, and *HMGA2* [68]. Based on this knowledge, a study attempted to overcome chemoresistance by co-delivering let-7a and doxorubicin using NPs composed of a magnetic core and mesoporous silica. They reported higher tumor growth inhibition in mice bearing breast cancer cells injected with both NPs co-delivering let-7a and doxorubicin compared to those treated with NPs delivering let-7a or doxorubicin alone [69]. Additionally, circulating miRNAs, which are detectable in biopsies and blood, have emerged as diagnostic biomarkers for cancer detection and evaluation of the response to cancer treatment [70]. There are few diagnostic tools based on miRNA detection available on the market [71].

In the context of fertility preservation, miRNAs are gaining attention due to their endogenous nature and their ability to target multiple signaling pathways, in contrast with the other aforementioned pharmacoprotective agents. Since cancer treatments can induce damage to the ovaries at different biological levels, the ability of miRNAs to target multiple factors is a unique and advantageous feature. Furthermore, miRNAs are known to play a role in ovarian function, including follicular development, oocyte maturation, and steroidogenesis, in normal physiology but also in metabolic and gynecological diseases, as extensively detailed in a review by Alexandri et al. [21]. However, it is only recently that miRNA-based therapy has been considered as a potential future fertility preservation method. In a study from 2016, miR-10a transfected using liposomes demonstrated reduced apoptotic effects on mouse granulosa cells exposed to nitrogen mustard in vitro. Additionally, this study demonstrated that miR-10a prevented follicles from undergoing atresia after mice were injected with busulfan, a well-known gonadotoxic alkylating agent. The mechanism of action appeared to involve BIM, a regulator of apoptosis targeted by miR-10a [72]. However, another study also focused on miR-10a to prevent gonadotoxicity induced by 4-hydroperoxycyclophosphamide (4-HC), the active metabolite of another alkylating agent (cyclophosphamide), and did not show a rescue of damage using liposomes to deliver the miRNAs in post-natal day 3 (PND3) ovaries in vitro [73]. However, the same research team analyzed the expression profile of 384 characterized miRNAs in ovaries before and after 4-HC exposure in vitro and identified let-7a as a promising candidate for preventing apoptosis of primordial follicles induced by chemotherapy. They reported a reduced apoptotic effect on PND3 ovaries when transfected with mimic-let-7a delivered by liposomes [74]. Moreover, they evaluated oocyte competence using in vitro-treated PND3 ovaries transplanted under the kidney capsule of adult mice to observe further in vivo follicular development. An improvement in follicular survival and oocyte quality was observed in the PND3 ovaries transfected with mimic-let-7a and exposed to 4-HC in vitro compared to the group exposed to chemotherapy alone prior to transplantation [75]. Let-7a targets genes involved in the cell cycle and apoptosis, as already mentioned. Other studies evaluated the protective effect of miRNAs against gonadotoxicity induced by chemotherapy in rats by targeting *PTEN* using either miR-21 [76,77], miR-144-5p [78], or unidentified miRNAs [79], all through mesenchymal stem cell-derived exosomes. However, PTEN is a negative regulator of the PI3K/Akt pathway involved in follicular activation. Inhibiting *PTEN* could lead to the depletion of ovarian reserve by “over-activating” quiescent follicles through the activation of the PI3K/Akt pathway. Another study focused on miR-144, but its 3p strand showed only partial protection of primordial follicles damaged by cisplatin exposure in adult mouse ovaries in vivo. The mechanism of action was suggested to involve the targeting of *MAP3K9* by miR-144-3p, which regulates apoptosis via the p38 MAPK pathway [80].

## 4. Targeting the Ovaries

The challenge of actively targeting specific tissues lies in the identification of the most appropriate markers, which should not only be specific to the tissue of interest but also be present on the surface of the most accessible cell type, and more precisely here, under normal physiological conditions. This specificity is more challenging than targeting tumor cells, which tend to overexpress proteins, including membrane proteins, due to their particular profiles that distinguish them from healthy cells. Moreover, tumors are highly vascularized, which induces the previously mentioned EPR effect, making it easier to target the tissue. Consequently, much of this research on targeting the ovaries has focused on ovarian cancer cells. This focus is also driven by the broader applications of ovarian cancer research, given the significant interest in nanomedicine for cancer and the accessibility of ovarian cancer cell lines, thanks to their natural immortality features. Hence, ovarian cancer cells can overexpress surface markers such as HER2 and folate receptors. These overexpressed receptors can be exploited for targeted drug delivery to enhance the efficacy of chemotherapy. For instance, one study utilized trastuzumab, an antibody targeting HER-2, to deliver cisplatin loaded onto poly(lactic-co-glycolic acid) (PLGA) NPs. The results showed higher cytotoxicity in SKOV-3, a human ovarian cancer cell line, compared to free cisplatin, and better internalization in cells with HER2 receptors compared to HER2-negative cell lines [81]. Another application involved targeting folate receptors using a nanocomplex system to deliver siRNA in SKOV3 cells [76]. GnRH, also known as luteinizing hormone-releasing hormone (LHRH), has its receptors overexpressed in ovarian cancer. In a study that employed AuNPs functionalized with an LHRH peptide, selective uptake of the AuNPs was observed in vitro. They also demonstrated, in vivo, preferential uptake of the AuNPs by organs in the abdominal cavity, primarily the ovaries, compared to other organs such as the liver, kidney, spleen, and pancreas [82]. Cancer stem cells (CSCs), which have the capacity for self-renewal, contribute to the heterogeneity of cancer cell populations. These cells, as stem cells, also express specific surface markers such as CD44 and CD133. These markers have been employed as targeting proteins for the delivery of drugs, such as paclitaxel, in mouse models [83]. Additionally, they have been used as proof-of-concept markers [84].

To effectively target the healthy ovary, it is crucial to understand the cell population within the tissue and identify the cell types that are most accessible through blood vessels. The ovaries consist of the cortex, where follicle development occurs, and the medulla, which is composed mainly of fibroelastic connective tissue and blood vessels. As the objective is to protect the female gametes from cancer therapy-induced damage, we will focus on follicle-specific markers, although the entire ovarian tissue should be protected from apoptosis. Ovarian follicles are the functional unit that contain the oocyte, surrounded by protective layer(s) of granulosa cells (GCs) and theca cells, both somatic cells [85]. The Human Protein Atlas, a database of human proteomes, provides open access to tissue-specific proteome maps, among other proteogenomic analyses [86,87]. The database reports 178 elevated genes and five enriched genes in the ovary, meaning that their expression is at least four-fold higher in the ovary compared to the tissue with the second highest mRNA expression level. Among these genes, only one encodes for a membrane protein, zona pellucida glycoprotein 4 (ZP4). This protein is specifically expressed by the oocyte and composes the zona pellucida, the extracellular matrix between oocytes and GCs. Recent research, driven by advancements in bioinformatic tools and technologies such as single-cell sequencing, has focused on the transcriptomics of human ovarian follicles. In a review by Zhang et al., human oocyte-specific markers such as *ZP1-2-3-4*, *DDX4*, *SYCP3*, *SOX30*, *ZAR1*, *DAZL*, *YBX2*, *H1FOO*, and *LHX8* were identified, along with GC-specific markers that included *CYP11A1*, *STAR*, *INHBA*, and *AMH* [88]. In another review by Chen et al. that studied novel regulators of follicle activation in mice, known oocyte-specific genes *BMP15*, *DAZL*, *DDX4*, *DPPA3*, *FIGLA*, *GDF9*, *LHX8*, *NOBOX*, *NPM2*, *OOG1*, *POU5F1*, *SOHLH1*, *SOX30*, *SUB1*, *SYCP3*, *TAF7L*, *YBX2*, *ZAR1*, *ZP2*, and GC-specific genes *AARD*, *ALDH1A2*, *AMH*, *AMHR2*, *CYP11A*, *CYP19A1*, *FOXL2*, *FSHR*, *FST*, *GATM*, *GNG13*, *HMGCS2*, *INHA*, *INHBA*, *KITL*, *KRT8*, *KRT19*, *RSPO1*, *STAR*, *UPK3B*, and *WNT6* were used to score single-cell gene expression [89]. Considering that many of the genes mentioned in these reviews are involved in molecular pathways, they may not be the best candidates as markers for targeting the ovary. Notably, gonadotrophin receptors, *FSHR* and *LHCGR*, are known to be expressed on ovarian follicles and are listed as mostly expressed in the ovary and testis in the NCBI protein-coding gene database. Another review that focused on oocyte human transcriptomes reported that *TGFBR1-2* and *BMPR2* were expressed at the membrane of oocytes [90]. While spatial proteomic studies are necessary to determine how to exploit this list of cell-specific genes, the known specific receptors are promising (Table 3). For instance, the conjugation of the LHRH peptide to AuNPs has shown potential in cancer research and could be extended to other applications if similar results are observed in healthy mice.

## 5. Limitations and Perspectives

In this review, we have addressed the question of how nanotechnology can offer new perspectives for future fertility preservation strategies and expand restoration options. With breakthroughs in nanomedicine and innovative therapeutic approaches for cancer treatments, including miRNA-based therapy, research into next-generation delivery systems is gaining momentum. We have illustrated how AuNPs can meet expectations as a promising vector for transporting biomolecules and targeting specific tissues by examining the most advanced studies utilizing AuNPs in diagnostics, treatments, and other applications. As previously mentioned, AuNPs are remarkable in terms of biocompatibility, stability, tunability, functionality, and scalability due to the ease with which they can be synthesized. However, it is important to recognize that the size, shape, and charge of AuNPs, resulting from the chemical surface composition, play a crucial role in determining their toxicity and pharmacodynamics. These properties can vary significantly between studies, making it challenging to summarize the best functionalization strategy, which must be confirmed with each modification.

Furthermore, because of their great capacity to be modified (size, shape, charge) and functionalized (miRNAs, ligands, drugs), the lack of standardization in AuNPs has resulted in studies yielding conflicting results on the same research topic. For instance, size-dependent cytotoxicity may change based on biological parameters, such as the type of cell line studied [91] or the physiological moment when AuNPs are administered. In fact, a study revealed that the estrous cycle affects doxorubicin efficacy when delivered by LNPs to the ovary, with a higher accumulation during mouse ovulation. They also observed a size-dependent accumulation of AuNPs in the ovary [92]. The variability of these results can also be due to the various techniques used to detect cytotoxicity or uptake [93]. Therefore, the development of more standardized methods of synthesis and analysis could improve the reliability of results concerning the properties and behavior of AuNPs.

Another critical limitation of the application of AuNPs is their accumulation in non-target organs. The natural biodistribution of AuNPs in certain organs upon administration is inevitable but can be diminished using targeted ligands, as already discussed. Moreover, their efficient clearance from the target site once they have achieved their intended action is equally important. AuNPs are known to be non-biodegradable, and they can persist in the body for extended periods, as reported during the NU-0129 clinical study [32]. This issue becomes particularly relevant when the target organ is healthy, as opposed to cancerous tissue. However, a study has reported an unexpected intracellular biodegradation of AuNPs in primary human fibroblasts. They studied the biotransformation of AuNPs over a period of 6 months in vitro and observed a degradation of AuNPs by the lysosomes through oxidation of gold induced by ROS after two weeks of exposure to 4 nm AuNPs. This was followed by recrystallization through metallothionein, according to their transcriptomic study, forming structures resembling aurosomes [94]. Aurosomes are structures formed after the administration of gold salts, a treatment used historically to address conditions such as rheumatoid arthritis before the development of more effective solutions. If degraded AuNPs are recrystallized into aurosomes in some particular cases, this may mean that different forms of gold could share the same metabolic pathway, and the one utilized by gold salts has already been more extensively studied [95]. For instance, some studies have indicated the elimination of gold through urine several months after gold administration [96]. It should be noted that 7 nm and 12 nm AuNPs, also included in the study, showed longer degradation times, closer to several months rather than the two weeks for 4 nm AuNPs [94]. While these findings are promising, more extensive research is necessary to determine the specific conditions under which AuNP degradation and recrystallization occur. This could also determine the best route of administration for AuNPs in clinical applications. As listed in Table 1, the potential administration routes include oral administration, intravenous administration, and intradermal injection. As only a few clinical trials using different routes of AuNPs administration are ongoing, it is not yet possible to evaluate the most appropriate and safe one. Moreover, the route of administration depends on the application and considers the balance between the efficacy of the delivery of the cargo and the absence of an immune response. This last event is to be taken into serious consideration as a severe immune response can terminate a clinical trial, as with MRX34. Nevertheless, AuNPs also show remarkable anti-inflammatory properties [97]. Furthermore, as previously discussed, molecules able to improve immune tolerance can be functionalized into AuNPs.

MiRNA therapy also has its limitations. As for AuNPs, off-target effects can be limited with the help of target ligands conjugated to the delivery system. However, once miRNAs reach their intracellular site of action, their regulation of multiple target genes, which is viewed as a benefit compared to other molecules, could also lead to unanticipated biological responses. While bioinformatic tools have made significant strides in predicting miRNA target genes, these predictions remain speculative and must be validated through experimental studies. Additionally, the complex cellular environment, which is reactive to external and internal signals, cannot always be accurately simulated in in vitro models, which are not fully representative of physiological reality. Moreover, genetic diversity among human populations could explain the difficulty of translating preclinical findings to clinical applications, which has not yet been achieved almost systematically due to immune responses in patients. It is crucial to improve our knowledge of the impact on the immune system as well as the potential long-term effects of organ accumulation. It is important to mention this last point to patients enrolled in clinical studies regarding the ethical aspect so that they can fully understand the positive and negative issues of clinical research.

One key challenge in miRNA therapy is finding the right balance between toxicity and efficacy when determining the appropriate dose for administration. The ideal dose should reflect the physiological concentration of miRNAs lost in the disease state. In addition to efficacy, this consideration is important in order to not saturate the RISC complex and induce problems in other miRNA actions. Despite these limitations, it is crucial to remember that miRNAs were only discovered 30 years ago. Their potential for therapeutic use has been explored more recently, with the first miRNA therapy ending clinical phase I trials in 2012, just a decade ago. This rapid progress is noteworthy, especially considering that miRNA therapy research is still in its early stages of development in terms of historical timeframes.

## Figures and Tables

**Figure 1 ijms-24-16593-f001:**
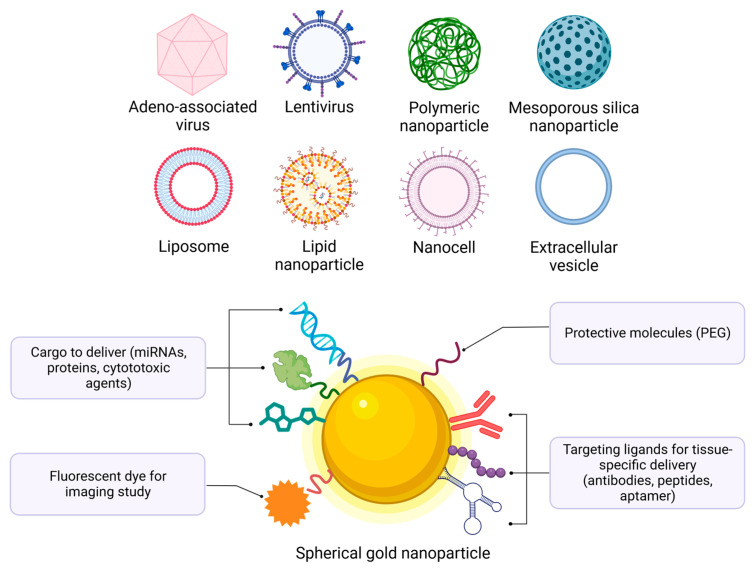
There are different types of delivery systems: virus vectors, polymeric-based, lipid-based, inorganic-based, and extracellular vesicle-based vectors. Gold nanoparticle functionalization with various ligands, molecules, and cargo. This figure was created with BioRender.com (accessed on 21 September 2023) with the agreement number MU25VPRPQK.

**Figure 2 ijms-24-16593-f002:**
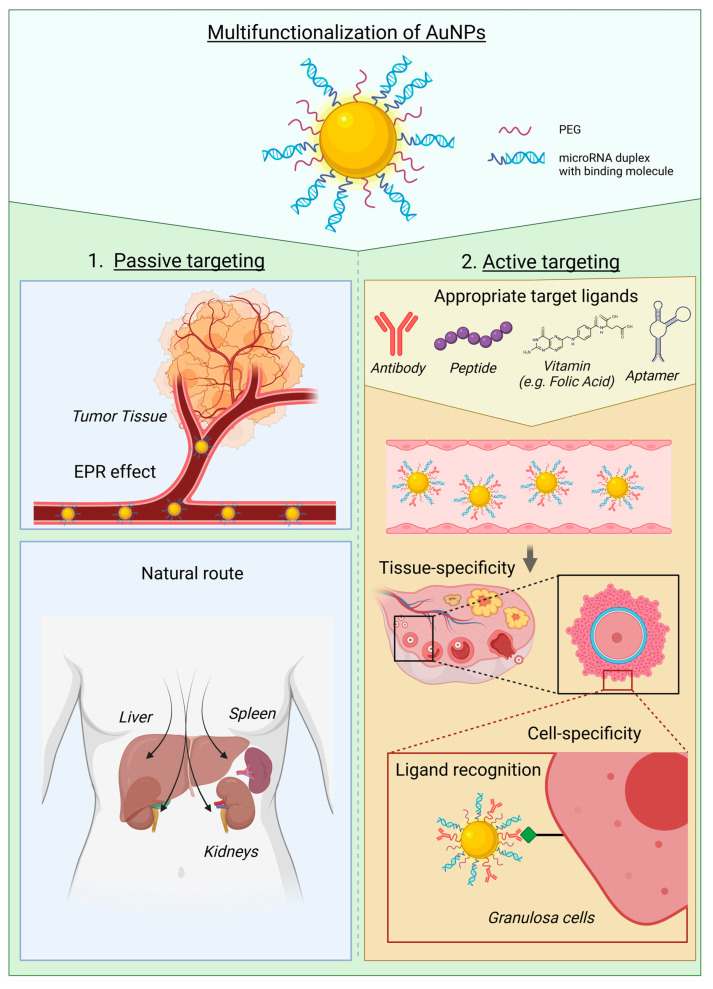
Gold nanoparticles are conjugated to specific target ligands, microRNAs, and PEG. There are two types of targeting phenomena: (1) Passive targeting of AuNPs exploiting the EPR effect of the tumor or natural route after administration; (2) Active targeting of AuNPs requiring functionalization with target ligands such as antibodies, peptides, vitamins (e.g., folic acid), or aptamers (DNA, RNA, or peptide molecules that bind to a specific target molecule). Once the appropriate target ligand is selected, AuNPs can reach specific cells by utilizing receptor-ligand recognition. AuNP: Gold nanoparticle; EPR: Enhanced permeability and retention; PEG: Polyethylene glycol. This figure was created with BioRender.com (accessed on 21 September 2023) with the agreement number WN25VPRLH5.

**Figure 3 ijms-24-16593-f003:**
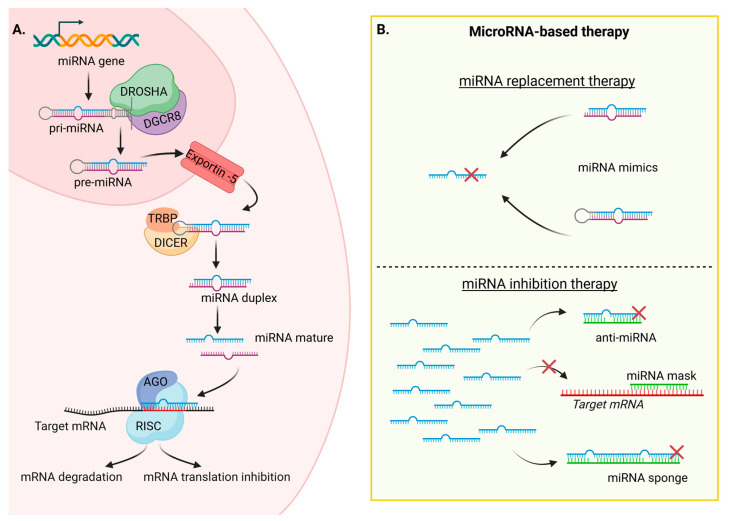
(**A**) MicroRNA biogenesis starts with the transcription of pri-miRNAs in the nucleus. The pri-miRNAs are processed into pre-miRNAs and into mature miRNAs. Finally, miRNAs can act at the post-transcriptional level to inhibit target mRNAs. (**B**) MicroRNA-based therapy: miRNA replacement to rescue downregulated miRNAs using mimic miRNAs and miRNA inhibition therapy to inhibit overexpressed miRNAs using anti-miR, miRNA mask, or miR sponge. AGO: Argonaute, DICER: Double-stranded RNA-specific endoribonuclease, DGCR8: DiGeorge syndrome critical region 8, DROSHA: Class 2 ribonuclease III enzyme, miRNA: microRNA; pre-miRNA: precursor microRNA; pri-miRNA: primary microRNA; RISC: RNA-induced silencing complex; TRBP: Transactivation response RNA binding protein. This figure was created with BioRender.com (accessed on 21 September 2023) with the agreement number DE25VPRBWM.

**Table 1 ijms-24-16593-t001:** Gold nanoparticles as a drug delivery system in clinical trials.

Name	Molecule	AuNPs	Administration Route	Treatment	Clinical Phase
CYT-6091	rhTNF	27 nm PEGylated	Isolation perfusion at the extremity	Various solid tumor	Phase I (NCT00356980) 2006–2009
C19-A3	Proinsulin peptide	5 nm	Microneedle injection	Type I diabetes	Phase I (NCT02837094) 2016-
Nano Swarna Bhasma	Combination of phytochemicals	35 nm	Oral capsule	Breast cancer	Phase 0 DNA_SPN_B001_17 AYUSH
NU-0129	RNAi for *Bcl2L12*	13 nm Thiolated PEG	Intravenous injection	Gliobastoma	Phase 0 (NCT03020017) 2017–2020
DengueTcP (EMX-001)	Synthetic T cell-selective multivalent with dengue virus peptide antigens vaccine	5 nm	Intradermal injection	Dengue fever	Phase I (NCT04935801) 2021-
Corona TcP	Betacoronavirus T cell-priming immune vaccine	5 nm	Intradermal injection	SARS-CoV-2	Phase I (NCT05113862) 2022-

rhTNF: recombinant human tumor necrosis factor, *Bcl2L12*: Bcl2Like12, PEG: Polyethylene glycol, SARS-CoV-2: Severe acute respiratory syndrome coronavirus 2.

**Table 2 ijms-24-16593-t002:** Micro-based therapy in clinical trials.

Name	Molecule	Delivery System	Treatment	Target	Clinical Phase
Miravirsen (RG101)	Anti-miR-122	LNA-antisense	Chronic hepatitis C	Liver	Phase II (NCT01727934) 2012–2014 Unknown
MRX34	miR-34 mimic	LNPs	Advanced solid tumors	Tumor	Phase I (NCT02862145) 2016–2017 Withdrawn
MesomiR-1	miR-16 mimic	EnGeneIC Dream Vectors	Malignant pleural mesothelioma	Tumor expressing EGFR	Phase I (NCT02369198) 2014–2017 Completed
Lademirsen (RG-012)	Anti-miR-21	Oligonucleotides modification	Alport Syndrome	Kidney	Phase II Suspended
Cobomarsen (MRG-106)	Anti-miR-155	LNA-antisense	Mycosis fungoides	Skin	Phase II (NCT03713320) 2019–2020 Terminated ^1^
			TLV-associated adult T-cell lymphoma/leukemia, diffuse large B-cell lymphoma, and chronic lymphocytic leukemia	Lymphatic system	Phase I (NCT02580552) 2016–2020 Completed
Remlarsen (MRG-201)	miR-29 mimic	LNA-mimic	Fibrotic diseases		Phase II (NCT03601052) 2018–2020 Completed
Obefazimob (ABX464)	miR-124 mimic	Capsule (oral administration)	Active Rheumatoid Arthritis	Immune system	Phase II (NCT05177835) 2021- Recruiting
			Ulcerative Colitis		Phase III (NCT05507203) 2022- Recruiting

^1^ Terminated for business reasons. LNA: Locked nucleic acid; LNP: Lipid nanoparticle, EGFR: Epidermal growth factor receptor.

**Table 3 ijms-24-16593-t003:** Ovarian markers based on human and mouse transcriptomic studies.

Name	Abbreviation	Proposed Cell Specificity	Model	Ovarian RNA Expression ^1^	Location
*Alanine and Arginine-rich* *Domain Containing Protein*	*AARD*	GCs	Human	N/M	Intracellular
*Aldehyde Dehydrogenase 1* *Family Member A2*	*ALDH1A2*	GCs	Human	Endometrial stromal cells	Intracellular
*Anti-Müllerian Hormone*	*AMH*	GCs	Mouse and human	GCs	Secreted
*Anti-Müllerian Hormone* *Receptor Type 2*	*AMHR2*	GCs	Human	GCs	Membrane, Intracellular
*Bone Morphogenetic* *Protein 15*	*BMP15*	Oocytes	Human	N/M	Secreted
*Bone Morphogenetic* *Protein Receptor Type 2*	*BMPR2*	Oocytes	Human	N/M	Membrane
*Cytochrome P450 Family 11* *Subfamily A Member 1*	*CYP11A1*	GCs	Mouse and human	N/M	Intracellular
*Cytochrome P450 Family 19* *Subfamily A Member 1*	*CYP19A1*	GCs	Human	N/M	Membrane, Intracellular
*Deleted in Azoospermia Like*	*DAZL*	Oocytes	Mouse and human	Oocytes	Intracellular
*DEAD-Box Helicase 4*	*DDX4*	Oocytes	Mouse and human	Oocytes	Intracellular
*Developmental Pluripotency* *Associated 3*	*DPPA3*	Oocytes	Human	Oocytes	Intracellular
*Folliculogenesis Specific* *Bhlh Transcription Factor*	*FIGLA*	Oocytes	Human	Oocytes	Intracellular
*Forkhead Box L2*	*FOXL2*	GCs	Human	GCs, Ovarian stromal cells, Endometrial stromal cells	Intracellular
*Follicle Stimulating* *Hormone Receptor*	*FSHR*	GCs	Human	GCs	Membrane, Intracellular
*Follistatin*	*FST*	GCs	Human	GCs	Secreted, Intracellular
*Glycine Amidinotransferase*	*GATM*	GCs	Human	GCs	Intracellular
*Growth Differentiation Factor 9*	*GDF9*	Oocytes	Human	Oocytes	Secreted, Intracellular
*G Protein Subunit Gamma 13*	*GNG13*	GCs	Human	N/M	Intracellular
*H1.8 Linker Histone*	*H1FOO*	Oocytes GCs	Human Mouse	Oocytes	Intracellular
*3-Hydroxy-3-Methylglutaryl-* *CoA Synthase 2*	*HMGCS2*	GCs	Human	N/M	Intracellular
*Inhibin Subunit Alpha*	*INHA*	GCs	Mouse and human	GCs Ovarian stromal cells	Secreted
*Inhibin Subunit Beta A*	*INHBA*	GCs	Mouse and human	N/M	Secreted
*KIT Ligand*	*KITL*	GCs	Human	N/M	Membrane, Intracellular
*Keratin 8*	*KRT8*	GCs	Human	N/M	Intracellular
*Keratin 19*	*KRT19*	GCs	Human	N/M	Intracellular
*Luteinizing Hormone/* *Choriogonadotropin Receptor*	*LHCGR*	GCs	Human	Ovarian stromal cells	Membrane, Intracellular
*LIM Homeobox 8*	*LHX8*	Oocytes	Mouse and human	Oocytes	Intracellular
*NOBOX Oogenesis Homeobox*	*NOBOX*	Oocytes	Human	Oocytes	Intracellular
*Nucleophosmin/* *Nucleoplasmin 2*	*NPM2*	Oocytes	Human	Oocytes	Intracellular
*Oocyte-Specific Gene*	*OOG1*	Oocytes	Human	Oocytes	Intracellular
*POU Class 5 Homeobox 1*	*POU5F1*	Oocytes	Human	N/M	Intracellular
*RNA Polymerase I Subunit A*	*RPO1*	GCs	Human	N/M	Intracellular
*Spermatogenesis and* *Oogenesis Specific Basic* *Helix-Loop-Helix 1*	*SOHLH1*	Oocytes	Human	Oocytes	Intracellular
*SRY-Box* *Transcription Factor 30*	*SOX30*	Oocytes	Mouse and human	N/M	Intracellular
*Steroidogenic Acute* *Regulatory Protein*	*STAR*	GCs	Mouse and human	Ovarian stromal cells	Intracellular
*SUB1 Regulator of* *Transcription*	*SUB1*	Oocytes	Human	N/M	Intracellular
*Synaptonemal Complex* *Protein 3*	*SYCP3*	Oocytes	Mouse and human	Oocytes	Intracellular
*TATA-Box Binding Protein-Associated Factor 7 Like*	*TAF7L*	Oocytes	Human	N/M	Intracellular
*Transforming growth* *factor beta receptor 1*	*TGFBR1*	Oocytes	Human	N/M	
*Uroplakin 3B*	*UPK3B*	GCs	Human	N/M	Membrane, Intracellular
*Y-Box Binding Protein 2*	*YBX2*	Oocytes	Mouse and human	N/M	Intracellular
*Zygote Arrest 1*	*ZAR1*	Oocytes	Mouse and human	Oocytes	Intracellular
*Zona Pellucida* *Glycoprotein 2*	*ZP-2*	Oocytes	Mouse and human	N/M	Secreted, Membrane

^1^ The RNA specificity category is based on mRNA expression levels in the analyzed cell types based on scRNA-seq data from normal tissues on the Human Protein Atlas. GCs: Granulosa cells, N/M: Not mentioned.

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
