# Peer review of "A Journey to Reach the Ovary Using Next-Generation Technologies"

_ijms, 2023, doi:10.3390/ijms242316593_

Round 1

Reviewer 1 Report

Comments and Suggestions for Authors

The article " A Journey To Reach Ovary Using Next-Generation Technologies" by Nguyen et al is devoted to focus on the advances in pharmaco protective approaches and the challenge of targeting the ovaries to deliver therapeutic agents. They have thoroughly discussed how AuNPs meet many of the requirements for an ideal drug delivery system, as well as the existing limitations that have hindering the progression of AuNPs research into more clinical trials. Additionally, the review highlights microRNA (miRNA) therapy as a next-generation approach to address on the context of fertility preservation and discusses the obstacles that currently impede its clinical availability.

The manuscript is truly attractive, and I strongly recommend accepting publishing this article in its current form.  

Author Response

Thank you very much for these positive comments.

Kindly find in attachment the R1 version of the manuscript, revised for improving English, and including some minor modifications.

Reviewer 2 Report

Comments and Suggestions for Authors

In the current study the authors explored how gold nanoparticles (AuNPs) can represent strong candidates for drug delivery of protective molecules into the ovary, particularly miRNA therapy.

The manuscript is clear, well written and describes these approaches in details.

However, I suggests the authors to further discuss these points.

1. Way of AuNPs administrations and their effectiveness

2. Putative role of the immune response and mechanisms of immunotolerance

3. Any ethical and/or legal issues regarding linical trial

Comments on the Quality of English Language

Minor editing of English language required

Author Response

We would like to thank the reviewer for their constructive comments.

A medical writer, Sandy Field,  has completely revised the manuscript for English language editing. Kindly find here the specific modifications added accordingling to your comments:

1. Way of AuNPs administrations and their effectiveness

- Additional column about Administration Routes in Table 1

- Additional discussion from lines 561 to 567

2. Putative role of the immune response and mechanisms of immunotolerance

- Additional discussion from lines 568 to 570

3. Any ethical and/or legal issues regarding clinical trial

  • Additional discussion from lines 582 to 585
